# Traumatic Brain Injury: Ultrastructural Features in Neuronal Ferroptosis, Glial Cell Activation and Polarization, and Blood–Brain Barrier Breakdown

**DOI:** 10.3390/cells10051009

**Published:** 2021-04-24

**Authors:** Delong Qin, Junmin Wang, Anh Le, Tom J. Wang, Xuemei Chen, Jian Wang

**Affiliations:** 1Department of Human Anatomy, College of Basic Medical Sciences, Zhengzhou University, Zhengzhou 450001, China; dragon@stu.zzu.edu.cn (D.Q.); wangjunmin@zzu.edu.cn (J.W.); xxchxm@163.com (X.C.); 2Program in the McKelvey School of Engineering, Washington University in St. Louis, Saint Louis, MO 63130, USA; anhsle15@gmail.com; 3Winston Churchill High School, Potomac, MD 20854, USA; twangy64@gmail.com

**Keywords:** cell structure, ferroptosis, inflammation, mitochondria, neuron, TBI, ultrastructure

## Abstract

The secondary injury process after traumatic brain injury (TBI) results in motor dysfunction, cognitive and emotional impairment, and poor outcomes. These injury cascades include excitotoxic injury, mitochondrial dysfunction, oxidative stress, ion imbalance, inflammation, and increased vascular permeability. Electron microscopy is an irreplaceable tool to understand the complex pathogenesis of TBI as the secondary injury is usually accompanied by a series of pathologic changes at the ultra-micro level of the brain cells. These changes include the ultrastructural changes in different parts of the neurons (cell body, axon, and synapses), glial cells, and blood–brain barrier, etc. In view of the current difficulties in the treatment of TBI, identifying the changes in subcellular structures can help us better understand the complex pathologic cascade reactions after TBI and improve clinical diagnosis and treatment. The purpose of this review is to summarize and discuss the ultrastructural changes related to neurons (e.g., condensed mitochondrial membrane in ferroptosis), glial cells, and blood–brain barrier in the existing reports of TBI, to deepen the in-depth study of TBI pathomechanism, hoping to provide a future research direction of pathogenesis and treatment, with the ultimate aim of improving the prognosis of patients with TBI.

## 1. Introduction

Traumatic brain injury (TBI) is and continues to be a major problem around the world. TBI refers to the brain tissue damage caused by external mechanical forces. Various disasters such as falling, traffic accidents, and violence can induce brain tissue damage and neuronal cell death present in TBI. TBI is one of the main causes of death and disability among children and adolescents, and approximately 69 million people worldwide are suffering from TBI each year [1,2]. Among these, approximately 1.5 million people in the United States and 3–4 million people in China experience TBI every year [3,4]. The care of TBI costs approximately $400 billion annually, representing about 0.5% of the entire annual gross world product [5].

There is a long way to go to solve the massive clinical problems caused by TBI. After TBI, patients may have brain edema, inflammation, nerve injury, motor deficits, cognitive impairment, and other signs and symptoms, which are capable of causing disability and death. It is estimated that TBI is going to be the 4th leading cause of disability-adjusted life years in 2030 [6]. TBI represents about 30–40% of all injury-related deaths across all ages, and we expect the same trend in disability rate until 2030 [5]. In recent years, researchers have revealed many pathways involved in TBI injury and elucidated the pathologic mechanisms underlying clinical symptoms after TBI. However, the pathomechanism of TBI is very complex, with many unexplored mechanistic pathways. Although many promising therapies and drug candidates can improve TBI outcomes in animals, no clinical trials have been successful so far. The treatment for severe TBI patients is essentially symptomatic and life-sustaining. Acute TBI is characterized by primary and secondary injuries. Primary brain injury refers to the direct injury to brain parenchyma during the initial impact. According to the biomechanical characteristics of impact, it can be focal or diffuse. Secondary brain injury is extensive and lasting which is caused by a multifactorial set of events including glutamate excitotoxicity, perturbation of cellular calcium homeostasis, membrane depolarization, mitochondrial dysfunction, inflammation, increased free radical generation and lipid peroxidation, neuronal death, and diffuse axonal injury [7]. During this process, a variety of pathways including excitotoxic injury, mitochondrial dysfunction, oxidative stress, ion imbalance, inflammation, and increased vascular permeability are activated. The serious short and long-term sequelae of TBI results from direct primary and complicated secondary injury. The variety of biochemical and molecular signaling connections involved contributes to the pathology of TBI, and these complicated connections are the main obstacle in TBI research, limiting the progress of drug development and clinical practice. Using electron microscopy, we can investigate the detailed structure of tissues, cells, organelles, and macromolecular complexes, which help us understand the structural basis of cell dysfunction after TBI. 

The purpose of this review is to summarize and discuss the ultrastructural changes in neurons, glial cells, and blood–brain barrier based on the existing literature of TBI, so as to move TBI research forward, hoping to provide a direction for future research in diagnosis, treatment, and prognosis of TBI in clinic.

## 2. TBI-Related Ultrastructural Damage to Neurons

### 2.1. Hydropic Disintegration of Neuronal Body and Process 

Regarding normal neurons, the extracellular space is small, and the myelin sheath is dense and well preserved. The neuronal cell body contains a large number of cytoplasmic organelles (such as Golgi body, rough endoplasmic reticulum, intermediate filament). The nucleus contains a small amount of heterochromatin and exhibits a complete nuclear membrane around it. In addition, the expected complement of intermediate filament and microtubule are obviously observed in a single cytoplasmic process, and the synapse is easy to be identified [1]. Establishing a TBI model in mice by controlled cortical impact (CCI) and using a new technique of continuous brain sectioning, Wiley CA et al. observed the cell bodies and processes of nerve cells in the injured mouse brain under electron microscopy [1]. These cell bodies and cell processes show water-soluble disintegration (or cerebral edema) under electron microscopy. This structural change includes loss of nuclear heterochromatin, and a small number of heterochromatin components in the cytoplasm; loss of microfilaments, microtubules, and mitochondria; increased number of electron dense film structure; and a large number of nuclear membrane fragments scattered throughout the whole cell body and neurite [1]. At the same time as water sample disintegration of cell body and protuberance, the cell membrane of these cells remained intact within 3 weeks after injury. In the contralateral cerebellar cortex, the nucleus of Purkinje neurons was sparse, and the cytoplasm was water-soluble disintegration and mainly manifested as subcellular apparatus dissolution, microfilament and microtubule loss, and the existence of electron-dense membrane fragments [1].

Cerebral edema (CE) is defined as an increase in brain water content in the affected tissue, including in individual cells and their surrounding interstitial space, and the excess accumulation of fluid within the brain. If unchecked, CE can lead to an increased intracranial pressure and fatal brainstem herniation. In fact, CE accounts for 50% of deaths in severe head injuries [8]. After TBI, CE forms at the lesion and incorporates into the surrounding tissue [9]. At present, the research on CE caused by TBI has made rapid progress, but the relevant mechanism is still unclear. Cellular edema can occur in all brain cell types, including astrocytes, endothelial cells, and neurons, and it has been a feature in astrocytes [10]. Several signaling molecules underlying edema formation in TBI, including Na^+^-K^+^-2Cl^−^ cotransporter [11,12], Aquaporins [13], Sulfonylurea-receptor 1–transient receptor potential member 4 (Sur1- Trpm4) [14,15], Glutamate [10,16], and Arginine vasopressin [17,18] have been studied in recent years, but the exact mechanism is still unclear. Continuous research is required to gain a thorough understanding. It should be noted that treatment strategies with an aim to mitigate CE after it has been formed may be less promising than those with an aim to inhibit the signaling pathways that contribute to edema formation [8].

### 2.2. Axon Destruction, Demyelination, and Myelination

Pedachenko et al. conducted an experimental TBI study with a weight-drop model, in which a weight (450 g) is dropped from a 1.5 m height to induce TBI [19]. Twenty-five adult (6-8 months old) and twenty elderly (24 months old) male rats were tested in the study [19]. They observed that after TBI, there were ultrastructural changes not only around the nucleolus but also in the processes of neurons and more than 1/3 of myelin sheath fibers were damaged [19]. A typical change following TBI includes axonal destruction with local swelling and axial cylinder deformation. In some cases, the axon was separated from the myelin sheath (vacuoles formed between the sheath and the axon) combining with disintegration, homogenization, and destruction of the neurite structure within the cell (Figure 1) [19]. In addition, the myelin sheath had disintegrated, the lamellar structure was destroyed, and swelling and protuberance had formed [19]. Mierzwa AJ et al. performed a mouse TBI experiment in which the corpus callosum was damaged [20]. At each time point of examination from 3 days to 6 weeks post TBI (3rd day, 1st week, 2nd week, and 6th week), the electron microscope showed that there were degenerated axons distributed in the intact fibers of the corpus callosum. Additionally, the degenerated axons increased significantly with time after TBI. The number of degenerated axons increased 2.4-3.8 times after TBI compared with the baseline value in sham mice. Six weeks after TBI, the axon degeneration increased by 2.9 times. These results demonstrated a delayed pathologic response after TBI. In a blast-induced mild TBI model, the clear myelin sheath ballooning was observed [21]. Song H et al. used open-field low-intensity blast (LIB) to study TBI [22]. The 3-m LIB-exposed animals exhibited clear myelin defects at 7 days post injury (DPI). They presented with extensive split myelin layers, dense degeneration (pockets of dense cytoplasm and vacuole within the myelin layers), myelin ballooning (bulges produced by splits in the myelin sheath layers), myelin disruption (rupture of the myelin continuity), or myelin detachment (hypodense periaxonal space between the axon and the myelin sheath) [22]. In addition, Wiley CA et al. have observed that in the synaptic structure of the damaged cerebral cortex, the presynaptic contour is adjacent to the axons of water-like neurons that can distinguish the postsynaptic density. At the same time, no presynaptic vesicles were found in the axons of water-like neurons [1]. Under normal conditions, the outer space of nerve fibers is small [1]. The myelin sheath is dense and uniform [1]. There were complete mitochondria and normal intermediate filaments and tubules in the myelinated nerve fibers [1]. There was no significant change in the above structures in the neurons of the injured brain at 7 and 21 days after TBI, which was similar to the normal brain [1]. 

In the TBI experiment in mice by Mierzwa AJ et al., which damaged the corpus callosum [20], complete axons, degenerated axons, and lost axons (only myelin but not axons) were observed under the electron microscope (Figure 2). Normally, the amount of excessive myelin sheath around axons is minimal. However, after axonal injury, excessive myelin sheath around the injured axons increased significantly [20]. Compared with sham mice, the quantity of excessive myelin sheath increased significantly on the third day (62.6 times), in the first week (45.9 times), and in the second week (23.2 times) [20]. In fact, the amount remained significantly increased within 6 weeks [20]. Overall, myelination was positively correlated with the frequency of axon degeneration in TBI mice. The pathologic characteristics of axon and myelin sheath were observed in the brain cells of the corpus callosum at different time points after brain injury [20]. For example, the degenerated axon was scattered in the complete axon [20]. The myelin sheath was abnormal, including demyelination, myelin sheath collapsed, and over myelin sheath (the myelin sheath did not tightly cover the axon, so it was similar to “redundant myelin sheath”) [20]. The complete axon showed obvious demyelination in 3 days and demyelination in 1 week (Figure 3) [20]. At 3 days (1.8 times) and 6 weeks (1.5 times) after TBI, the number of demyelinated axons increased significantly, but only accounted for a small part of the total axons (<5%) [20]. The frequency of complete axon demyelination was not related to the existence of axonal degeneration. Remyelination refers to the fact that the myelin sheath is thinner than normal within a given axon diameter. 

In a TBI study by Mierzwa AJ et al. [20], compared with the sham group, the thickness of myelin sheath of the intact fibers decreased at 1, 2, and 6 weeks. Compared with the sham mice, the TBI mice showed a significant decrease in the average axon diameter at 3 days (10.2%) and 6 weeks (16.2%) [20]. The ratio of g (the formula is the axon diameter divided by myelinated fiber diameter) shows the existence of remyelination more specifically by calculating the size of the axon. The increase of the slope of g-ratio relative to the diameter of axon in the binocular indicates that the diameter of the axon with myelin is increased, which also indicates the progress of remyelination. From the first week, the remyelination in the whole fiber group showed not only the thinning of the myelin but also the increase of the slope of the G-ratio curve. Additionally, significant demyelination at 3 days returned to normal at 1 week, consistent with the progress of remyelination during this interval.

Although myelin is the main component of fibroblasts, the pathological effect of the myelin on TBI has not been elucidated. After axon loss, the related myelin sheath may collapse into double myelin sheath before degradation. In addition, in the process of biological formation of myelin sheath, myelin sheath can extend outward beyond the perimeter of the axon and then fold back to itself. These potential double myelin sheaths or lack of visible axons extending outward from the axons are classified as excessive myelin sheaths [20]. Among all the excessive myelin sheaths found, the number of time-span points of the excessive myelin sheaths with axon loss accounted for between 60.3% to 67.8%, and the excessive myelin sheaths were positively correlated with the increase of the frequency of axon degeneration in TBI mice. This indicated that a series of pathologic processes after TBI led to axon damage, and then led to different degrees of neurodegenerative diseases, such as cognitive impairment. After TBI, the affected axons exhibit obvious demyelination, indicating a series of pathologic processes after TBI lead to axon damage corresponding to clinical signs and symptoms. For example, the surviving axon demyelination damages the jump conduction, making the axon more vulnerable to further damage. The slow conduction along the demyelinated axon causes the nervous circuit to lose synchronization, and this may be one of the reasons for the decrease in information processing speed after TBI [23,24]. Myelin sheath damage may also lead to myelin fragments, stimulating the activation of microglia [25], thus leading to the generation or aggravation of neuroinflammation. 

During the development of the central nervous system (CNS), a part of the myelin formation was excessive myelination (with double sheath), which is similar to a “redundant” myelin sheath, and that was demonstrated by three-dimensional reconstruction analysis [26]. In previous preliminary studies, it was found that one of the significant characteristics of myelin pathology after injury was abnormal myelin sheath, over folding back [27]. Combined with the above experiments, there was continuous myelination of denatured axons within 6 weeks after TBI. These results indicate that the spontaneous myelination of surviving axons after demyelination may be particularly related to the recovery after brain injury. Moreover, research on the demyelinating disease model shows that remyelination can not only restore function, but also protect the demyelinating axon from progressive degeneration and disconnection, resulting in permanent loss of function [28,29]. To sum up, at any time point after TBI, the excessive myelin sheath in the corpus callosum is related to axonal degeneration, and most of the excessive myelin sheath is much longer than the expected collapse of the myelin sheath around the degenerated axon. These excessive myelin patterns may be caused by axon loss and/or myelin synthesis. In the developing central nervous system, the double myelin sheath is called “myelin sheath abduction” or redundant myelin sheath [26]. In the process of myelin biosynthesis, the excess myelin membrane extends radially from the growing axon, forming a loose layer, and gradually expands around the growing axon and other axons located near it [26]. The extension of these redundant myelin sheaths produces double myelin sheaths. Therefore, after TBI, excessive myelin sheath may reflect the synthesis of myelin sheath to regenerate the living axon and/or abnormal myelin sheath. That is, the remyelination is the main reason for the excessive morphology of the myelin sheath observed after TBI [20].

### 2.3. Mitochondrial Abnormality

Abnormal mitochondrial morphology is a key event in the development of neuronal damage, which is usually observed in TBI [30]. In the past few years, many researchers have found that TBI leads to damage to mitochondrial function and mitochondrial functional integrity [31]. At the same time, mitosis and fragmentation play a positive role in apoptosis cell death [32]. Mitochondrial dysfunction may trigger or aggravate secondary intracellular cascade damage after primary damage [31]. In addition, mitochondria participate in the endogenous apoptosis pathway, involving a variety of not receptor-mediated stimulation to generate intracellular signals [33], which may be related to a series of secondary injuries after TBI primary injury. In a study by Wu Q et al. who used the weight-drop model of TBI in mice, abnormal mitochondria in neurons were observed 24 h after TBI. These abnormalities included severe mitochondrial fragmentation, crista collapse, mitochondrial swelling, mitochondrial membrane rupture, decreased mitochondrial density, and increased size and shape heterogeneity (Figure 4) [34,35]. In addition, in the experiments by Wiley CA et al. who used the mouse CCI model, on the 7th and 21st day after TBI, the number and volume of mitochondria in the neurons in the damaged area decreased [1]. Pedachenko et al. conducted a TBI study with the weight-drop model of TBI in rats, in which a weight (450 g) is dropped from a 1.5 m height [19]. They observed that the mitochondria in the cytoplasm of most neurons in the damaged area showed destructive changes and rapidly expanded vacuoles with destruction and decomplexation of cristae and damaged membranes [19]. There was another obvious abnormal change in the mitochondria within the axon: crista disruption and obvious swelling [19]. Using a fluid percussion model of TBI in rats, Wang Q et al. observed that some cells in the injured area (cerebral cortex and hippocampus) had obvious ultrastructural damage on the 7th day after TBI, such as cell atrophy, mitochondrial swelling, reticulum expansion, and delayed hematoma [34]. In the blast-induced mild TBI mouse model by Song H et al., swollen mitochondria were observed [21]. In the LIB model of TBI in mice by Song H et al., extensive mitochondrial abnormalities within the neuropil were observed and characterized as swollen clear, swollen dense, and dark degenerated mitochondria [22].

Mitochondrial morphology and intracellular distribution depend on many factors, including mitochondrial energy state, membrane permeability change, physical interaction with the cytoskeleton, mitochondrial dynamics (motion, fission, fusion), and the balance between mitochondrial biogenesis and degradation [36,37,38,39,40]. Normal cells maintain a balance between mitochondrial fusion and division, thus maintaining normal mitochondrial morphology and aerobic metabolism. The morphological and structural abnormalities of mitochondria in different degrees after TBI indicated that the injury caused by external force resulted in the abnormality of mitochondria, and its normal physiological function was seriously affected. 

In the process of secondary injury after TBI, one of the most prominent consequences is the apoptosis of nerve cells. This apoptosis is closely related to the abnormalities described above in mitochondria. There are two pathways of apoptosis, one is the exogenous pathway mediated by specific receptors on the surface of the cell membrane, the other is the endogenous pathway involving mitochondria. When apoptosis begins, proteins called caspases are activated that break down various cell components required for cell survival [41]. Cytochrome c plays a key role in the process of apoptosis by translocating from mitochondria to cytoplasm and bins to apoptosis protease-activating factor 1 (Apaf-1) to induce apoptosome formation [42]. When injured, abnormal structural changes such as swelling, the collapse of crista, rupture of membrane, and so on occur in the mitochondria of nerve cells at the injured site. When the crista of mitochondria collapses and breaks, the cytochrome c in them will be released. Given that the majority of cytochrome c resides inside the narrow cristae junctions, the release of cytochrome c probably requires two steps: mobilization and translocation where the mobilization step may involve cristae remodeling [43]. Once released, cytochrome c, in interaction with the Apaf-1, triggers the initiator caspases-9 activation and then leads to the subsequent characteristic features of apoptosis, including chromatin condensation and nuclear fragmentation [44]. The cytochrome c binds with Apaf-1 and ATP to generate complex proteins called apoptosomes which cleave to procaspases that activate caspase 3 [45]. During apoptosis, mitochondria dramatically fragment as a consequence of increased recruitment of DRP1 to mitochondria, which is key to the positive regulatory role DRP1 plays in Bax/Bak-mediated mitochondrial outer-membrane permeabilization [46,47,48]. 

One of the other major causes of secondary injury from a TBI is an overproduction of radical oxygen species (ROS) [49]. Healthy mitochondria produce a small amount of ROS, but there is a set of antioxidant systems in the cell to protect it from the ROS attack at the same time. Under normal circumstances, ROS cause little damage as the balance between the generation and scavenging of ROS is highly controlled [50]. After a TBI, mitochondrial dysfunction can occur and release excessive ROS [51]. Changes in the normal reduction-oxidation (redox) state of cells can result in toxic effects through the production of peroxides and free radicals that damage all components of the cell, including proteins, lipids, RNA, and DNA [52]. When it comes to the secondary injury of TBI, the responses to oxidative stress involve changes in mitochondrial channels that can lead to ROS release, and these responses are called ROS-induced ROS release. With high levels of ROS, the mitochondrial permeability transition pore (mPTP) openings that maintain homeostasis of ROS within the cell are insufficient, and the mitochondria release an ROS burst, leading to the destruction of mitochondria [50]. Excessive ROS damages proteins, mtDNA, and lipids, resulting in apoptosis, neuroinflammation, and disruption of the blood–brain barrier (BBB) in TBI injured brains [53]. Excessive ROS induces axonal damage and apoptosis of oligodendrocytes through 4-Hydroxynonenal from lipid peroxidation [54]. To reduce cellular damage, the mitochondria will normalize ROS which usually occurs through brief mPTP openings [50]. In addition, the shape of the mitochondrial cristae regulates both mitochondrial efficiency and metabolic functions [55]. 

Under different physiological and pathological conditions, the ultrastructure and quantity of mitochondria are often quite different. Therefore, mitochondria can be used as a reliable sensitive index in cytopathology and play an important role in clinical examination. This can provide a method for the diagnosis, treatment, and prognosis of TBI patients.

After TBI, the physiologic environment of brain cells changes obviously with the progress of injury. This pathological change destroys the environmental factors that depend on the normal shape and distribution of mitochondria and finally leads to abnormal morphologic and structural changes such as mitochondrial fragmentation, crista collapse, mitochondrial swelling, mitochondrial membrane rupture, and mitochondrial density reduction.

### 2.4. Endoplasmic Reticulum Dissolution

In a mouse CCI model of TBI by Wiley CA et al., the damaged intracellular organelles were broken under electron microscope (as described in the following), and only a small amount of organelles were found in the cytoplasm (Figure 5) [1]. The rough endoplasmic reticulum was dissolved, and the residual was also broken [1]. The residual Golgi complex was difficult to identify [1]. The mitochondria were broken, lost, and the subjective volume was small [1]. In addition, a large number of membrane fragments were scattered in the whole cell body and neuron process [1]. Pedachenko et al. conducted a TBI experiment by dropping a 450 g weight from a 1.5 m height to induce TBI in male rats [19]. The Golgi cell vacuolation was observed in the injured cells following TBI [19]. Additionally, most mitochondria showed destructive changes and rapid expansion of vacuoles and some organelles were hypertrophic [19]. Moreover, the number and size of residual centrosomes in glial cells also increased [19]. In a recent weight-drop TBI model experiment of rats by Liu Q et al., a swelling endoplasmic reticulum was observed clearly [56].

Comprising between 15%–60% of the total cell membrane volume, the endoplasmic reticulum (ER) is an essential and evolutionarily specified cellular organelle involved in several processes, including protein homeostasis, stress response, survival signaling, and trafficking of secretory, as well as cell-surface proteins [57]. Additionally, the ER maintains a Ca^2+^ store and acts as a site for oxidative protein folding within cells [58]. The induction of a TBI not only causes direct damage, such as axon shearing, to tissue but also triggers a delayed sequence of cellular and molecular events that result in secondary injury, such as a disruption of Ca^2+^ homeostasis, which has been suspected as the fundamental pathological mechanism [59,60]. The dissolution of the endoplasmic reticulum after TBI is closely related to the destruction of calcium homeostasis. By serving as the major intracellular calcium (Ca^2+^) storage compartment, the ER plays a critical role in maintaining Ca^2+^ homeostasis among various cellular organelles [57,61]. As one of the most important intracellular signaling molecules in the control of proliferation, differentiation, secretion, contraction, metabolism, trafficking, and cell death, cytosolic Ca^2+^ is tightly regulated in time, space, and concentration [62]. Maintenance of balanced Ca^2+^ homeostasis is critical to all ER-supported physiologic functions [61]. Ca^2+^ accumulated within the ER lumen not only controls fast signaling events but also regulates numerous ER-residing chaperone enzymes in post-translational protein processing. The destruction of calcium homeostasis and the increase of unfolded or misfolded proteins caused by it lead to post-TBI ER stress [63]. ER stress stimulates the release of Ca^2+^ from ER to cytosol, and eventually leads to the release of cytochrome c and induces apoptosis [64]. Therefore, ER stress resulted from TBI modulates the mitochondrial apoptosis pathway via regulation of ER Ca^2+^ levels.

The unfolded protein response (UPR) is an intracellular stress response due to accumulation of misfolded and unfolded proteins in the ER [65]. It is a fundamentally adaptive cell response, which functions as a cytoprotective mechanism to overcome ER stress when ER homeostasis is perturbed by intraluminal Ca^2+^, infection, nutrient deprivation, improper glycosylation, accumulation of misfolded proteins, or changes in redox status [66,67]. However, if ER stress is prolonged and there is a sustained activation of the UPR, the cell’s propensity to combat ER stress is exhausted, resulting in the activation of pro-apoptotic pathways, such as the transcription and translation of C/EBP homologous protein and the activation of caspase-12, that subsequently eliminates cells injured by ER stress to ensure the survival of the organism [63,68]. 

There are three major ER stress sensor-proteins that are associated with the UPR: PKR-like ER kinase (PERK), inositol requiring kinase 1 (IRE1α), and activating transcription factor 6 (ATF6) [66]. These proteins mediate the alleviation of ER stress to realign protein-folding demand and capacity back into homeostasis so that the cell can survive and function [69]. Under normal physiologic conditions when the ER protein folding capacity corresponds to a load of newly synthesized proteins, the activity of these three ER sensor-proteins is suppressed by binding to an ER chaperone, a 78 kDa glucose-regulated protein (GRP78) [70]. However, as conditions of ER stress manifest through the accumulation of misfolded or unfolded proteins in the ER lumen, GRP78 dissociates from the ER stress-sensing proteins, thereby resulting in their activation. Subsequently, GRP78 binds to unfolded proteins to aid in the refolding process [70]. These three ER sensor-proteins act to alleviate ER stress, but if the stress is too severe or prolonged, programmed cell death will be triggered [66]. Direct damage to cellular proteins and folding mechanisms is also associated with TBI and further exacerbates aberrant protein accumulation and thus ER stress [71]. A disrupted, elevated ER Ca^2+^ homeostasis can also result in the activation of proteases, phospholipases, and the formation of oxygen and nitrogen free radicals [72].

Although the mechanism of these extensive ER stress responses is unclear, blocking ER stress might be a potential therapeutic option for TBI, which can not only reduce the accumulation of abnormal proteins and promote the recovery of neurons but also may decrease ER-associated apoptosis and promote neuroprotection.

### 2.5. Cytoskeleton Destruction

The cytoskeleton is an important structure commonly existing in eukaryotic cells. It refers to a protein fiber grid structure system consisting of microtubule (MT), microfilament (MF) and intermediate filament (IF) and plays a role of scaffolding in cell morphology and internal structure arrangement [73]. In the weight-drop model of TBI in rats by Pedachenko et al., it was observed that there were obvious changes in the cytoskeleton of rat nerve cells after TBI, such as the damage of nerve fibers and the local loss of microtubules, which also led to the interruption of axon transport [19]. The loss of microfilaments and microtubules was also observed in a mouse study by Clayton A. Wiley et al. [1]. In the mouse corpus callosum damaging TBI experiments by Mierzwa AJ et al., under the electron microscope, cells exhibited axonal degeneration, the cytoskeleton was broken, the cytoplasmic density was abnormal (51.1–59.9% was abnormally high, 23.7–33.1% was abnormally low), and there were vesicles in the cytoplasm [20].

At present, the mechanism of cytoskeleton destruction after TBI has not been fully elucidated. The cytoskeleton of submicron cells is composed of lattice-like tissues of various proteins, which helps to form and stabilize special domains [74,75]. In addition to the role of scaffolds, the cytoskeleton is also widely involved in material transport, cell movement, information transmission, gene expression, cell division, and differentiation. It is generally believed that the axon damage caused by TBI through axon shearing and other pathways is the cause of related neurological symptoms [59]. After the axon is transected in vivo, there is a latent period. The axon remains static and electrically excited in structure, followed by a rapid and irreversible process called granular disintegration of axon cytoskeleton (GDC), in which the neurofilament, microtubule, and other cytoskeleton components disintegrate [76,77]. It is generally believed that the local damage of the axonal outer membrane leads to the release of calcium, which leads to the fracture of the cytoskeleton and mitochondria and the connection between nerves. The destruction of the axon outer membrane allows the accumulation of calcium in the local axon, and then activates various calpain pathways, which can degrade the cytoskeleton network in the axon [78]. Calpain-mediated cytoskeleton degradation has been shown to occur in axon-damaged and disconnected sites and many immunohistochemical studies have used antibodies against their specific protein breakdown products [79]. 

Microtubules are important cytoskeleton components, whose structural integrity is the basis of axonal nutrient transport. The microtubules extend from the center of the centrosome to the surrounding area, providing a pathway for the transport of intracellular materials. Some transport vesicles, secretory granules, pigment granules, and other substances synthesized in cells are transported along the track provided by microtubules. The common disruption of anterograde and retrograde axonal transport after brain injury can be attributed to the depolymerization or loss of microtubules [79,80]. Although caspases play an important role in the death of apoptotic cells in axons with a severe injury, they are believed to be involved in the terminal degradation of the cytoskeleton, resulting in an irreversible collapse of the subperiosteal cytoskeleton rather than direct apoptosis of cell bodies [81]. The destruction of the cytoskeleton destroys axon transport and leads to the accumulation of organelles and vesicles, leading to axon swelling and final separation [82]. When microtubules are destroyed and lost, the transport of intracellular materials will be inhibited. The early disintegration or loss of microtubules may be due to the depolymerization of microtubule components, protein decomposition, or the combination of the two [83]. 

Neurofilament is the most abundant cytoskeleton protein in large myelinated axons, with longitudinal orientation and regular intervals [84]. Neurofilaments are the dominant intermediate filaments in axons and are produced in the cell bodies of neurons and transported throughout the axons. Structurally, they are specific heteropolymers assembled from a central rod-shaped domain surrounded by threefold proteins (which can be light, medium, or heavy) [85,86]. Neurofilament may be the key factor of axon tensile strength and mechanical tensile elasticity. Although the characteristics of single neurofilament segments have not been systematically examined, there is evidence that calcinosis of the neurofilament cytoskeleton leads to the loss of filamentary structure [87].

The study of cytoskeleton destruction after TBI is still in progress. At present, the detection of cytoskeleton damage can be used as a means to diagnose the severity of TBI or to observe its progress. In addition, it can be assumed that drug design targeting at blocking related pathways of cytoskeleton destruction will become a potential therapy for TBI.

## 3. Effects of TBI on Glial Cell Ultrastructure

### 3.1. Structural Changes of Astrocytes

In a TBI study in mice by Mierzwa AJ et al. in which the corpus callosum was damaged [20], astrocytes showed cytogenetic changes, indicating cytogenetic edema. The swelling of astrocytes was also clearly seen in the brain sections of rats 10 days after TBI. In a mouse CCI model of TBI, astrocytes in the damaged area and the surrounding area were hypertrophied 3 days after TBI. Seven days after injury, at the same time, the morphology was further changed and a glial scar was formed [88]. In another CCI model of TBI in mice by Susarla BT et al., GFAP-positive astrocytes proliferate on the 1st, 3rd, and 7th day after the injury, and the number of proliferating astrocytes reaches the peak on the 3rd day after injury. These astrocytes were located near the lesion, showing hypertrophy and prolonged protrusion (Figure 6) [89]. In addition, in TBI mice of another CCI model, obvious swelling of the foot process of astrocytes was observed by Yao X et al. [90]. In a test of biopsy specimens taken from the cerebral cortex of 18 patients with brain parenchymal injury by Shitaka et al., the electron microscopy showed swelling of astrocytic perivascular processes [91]. Cytotoxic edema of astrocyte end-feet was also observed at the same time in the same area [91].

Astrocytes are key players in the multicellular response to CNS trauma and disease [92]. Astrocytes play an important role in maintaining the physiological homeostasis of the central nervous system, supporting nerve function, and glial transmission and signaling through calcium release and absorption [93]. The interaction between astrocytes and endothelial cells is a key component of BBB induction and maintenance, which involves intercellular and intracellular communication [94]. Astrocytes also play a role in repairing BBB and maintaining homeostasis by providing metabolic support for neurons and their synapses [95,96]. It is well known that inflammatory response after brain injury leads to extensive cell death, chronic tissue degeneration, and functional disability [97,98]. Generally, astrocytes and microglia are considered to be the main inflammatory cells after various brain injuries. Astrocytes and microglia can secrete a variety of cytokines, chemokines, prostaglandins, and growth factors, and morphological changes will occur after the CNS is damaged [99,100]. Ultimately, these changes affect the local microenvironment, which determines the degree of damage, subsequent repair, and functional recovery [101]. 

How astrocytes transduce physical strain associated with more diffuse forms of tissue damage after TBI into subsequent changes in cell function is incompletely understood [95]. An increase in astrocyte reactivity in response to injury is termed astrogliosis [102] which involves changes in morphology, increased expression of the intermediate filament proteins, glial fibrillary acidic protein (GFAP) and vimentin, and heightened proliferation and secretion of inflammatory mediators and growth factors [103,104,105,106,107,108]. In this review paper, we focus on the morphological changes of reactive astrocytes. 

As mentioned above, the ultrastructural changes of astrocytes after TBI have taken place, including swelling of the cell body, swelling and the elongation of the foot process, and formation of a glial scab. TBI with severe focal tissue damage triggers inflammatory mechanisms essential for the clearance of debris [92]. In this process, astrocytes cooperate with phagocytic immune cells, releasing cytokines, chemokines, and inflammatory mediators (including high-mobility group box 1, heat shock proteins, and S100 proteins) to promote clearance of cytotoxic cellular debris and decrease inflammation [92]. The signal transduction of NF-κ-B in astrocytes mediated by pattern recognition receptors is an important cause of cell swelling, which is related to cytotoxic edema [109]. The swelling of astrocytes after TBI is considered to be one of the early signs of cytotoxic edema, which is supported by diffusion-weighted imaging measurement imaging research including closed brain injury, diffuse brain injury, weight loss, hydraulic impact injury, and controllable cortical impact injury [110,111,112]. High-mobility group box 1 released from TBI-induced tissue damage can signal microglia to secrete IL-6, and signal reactive astrocytes can up-regulate AQP4 water channel involved in the water absorption of astrocytes [113]. In response to local tissue damage or inflammation, reactive astrocytes form scar boundaries to separate damaged and inflamed tissue from adjacent potentially viable nerve tissue which are instrumental in regulating the propagation of tissue injury, inflammation, and instructing brain repair [92,102,114].

Astrocytes are usually one of the main types of cells that initiate the inflammatory cascade in the sense of danger. Proteins related to astrocyte activation are often used as biomarkers of TBI. Further study on the mechanisms of astrocytes in TBI can help us to better understand the complex secondary injury of TBI. A better understanding of the nature of the inflammatory response produced by astrocytes will help to develop therapies to fight cell death and degeneration and to protect brain tissues against TBI injury. It is a feasible direction to study the ultrastructural changes of astrocytes, which will be helpful to clinical diagnosis and to judge the severity and prognosis of TBI.

### 3.2. Structural Changes of Microglia

In an electromagnetic repetitive closed-skull TBI examination with 18 patients by Vajtr D et al. [115], in the affected brain regions examined, cells with the ultrastructural characteristics of activated microglia were observed near injured axons. These cells were often located near capillaries and had dense heterochromatin near the nuclear envelope, granular cytoplasm, and extended cytoplasmic processes. In some cases, these cytoplasmic processes were in direct contact with dystrophic axons. The activated microglia directly contact the damaged axons, suggesting that there is a certain relationship between the activation of microglia and axon damage [115]. Furthermore, in a mouse CCI model conducted by Kumar et al., as neuropathological and inflammatory changes were monitored within one year after experimental TBI, it was found that the microglia in the injured area was activated chronically, and the cell body was hypertrophic [116]. In a newly established closed head injury model of TBI in mice by Roth TL et al., microglia with elongated processes were observed around the meningeal cells that died from primary trauma [117]. Under the two-photon microscope, it is observed that microglia react by extending to the boundary of glial cells and limiting the foot processes of single astrocytes within a few minutes after brain injury, forming a structure similar to a hexagonal honeycomb [117]. In addition, when some cells died, microglia transformed into phagocytes, and the cell body formed a jellyfish-like structure [118]. 

Microglia are innate immune cells with phagocytosis and antigen presentation ability in the brain [119,120]. In a “resting” state, microglia have a rod-shaped cell body, and the process extends symmetrically in all directions [121]. Microglia activate rapidly after CNS injury. According to the nature of stimulation, microglia can present a variety of activation states, which correspond to the changes in microglial morphology, gene expression, and function [122,123]. As with other CNS injuries, microglial activation in TBI results in different phenotypes, corresponding to neurotoxicity or neuroprotective activation [123,124]. Depending on the stage and severity of the disease, microglia are stimulated differently, leading to specific activation states [122]. The classification of microglial activation has been controversial. The classical method of macrophage classification was borrowed and applied to microglia, and the activated state was divided into M1 (proinflammatory) state or M2 (anti-inflammatory) state [125,126]. The proinflammatory M1 phenotype is conducive to the production and release of cytokines, which can aggravate nerve damage [127]. After LPS or IFN-γ exposure, the cells differentiated from the “resting” or “M0” phenotype to the M1 phenotype, which is considered as neurotoxicity after CNS injury. After exposure to IL-4 or IL-13, the cells differentiate from the M2 phenotype [128,129]. In contrast, the M2 phenotype is associated with the release of neurotrophic factors that promote repair and phagocytosis [130,131]. 

After microglia activation, a series of characteristic morphological changes will take place [132]. In general, their foot processes move from undirected movement to targeted movement to the injured area [121]. Then, these pods begin to contract, and the cell body expands and becomes spherical [133]. Finally, microglia begin to migrate to the injured site at the rate of 1-2 μm per hour [121]. Microglia are usually present in a reactive state, but there is no phagocytosis residue in the cytoplasm. In TBI mice, microglia usually contain cell fragments that indicate phagocytic activity. The acute response of microglia to TBI is to remove cell and molecular fragments, which is an important step to restore normal brain homoeostasis [118]. It seems that the Iba1-immunoreactive cells with activated microglial morphology represent a response to primary-traumatic axonal injury. It is also possible that the axonal injury is secondary to toxic factors secreted by the activated microglia or that both the axonal injury and activated microglia are responses to some unknown pathologic factor [115]. The causal relationship between the two needs to be further researched. 

Generally, brain edema results from cytotoxic edema in neurons and usually leads to axon damage, such as axon degeneration, axon loss, and so on. These damages usually degrade the remaining myelin sheath and produce a lot of myelin fragments. These myelin fragments can stimulate the activation of microglia [25]. Therefore, too much myelin after TBI can lead to the occurrence of persistent neuroinflammation. In various brain injury models, the activation degree of microglia varies with the type and severity of the injury. The microglial receptor is a sensor that recognizes these subtle changes in the substance of neurons and leads to various activation patterns of microglia. Therefore, the expression pattern of a microglial receptor can be used as a marker of injury severity after TBI, and it may be a sensitive diagnostic tool [134]. Additionally, drug-targeted microglia-specific receptors will prevent the changes of neurophysiology, thus reducing unnecessary side effects. Therefore, microglial receptors may be an effective target for the treatment of TBI-induced brain injury [134].

The different responses of astrocytes and microglia in brain injury are due to their different stimulation from the surrounding cells and the local microenvironment [106,107,108,135,136,137]. Matrices, such as inflammatory mediators, proteases, complement factors, and damping, trigger complex cascade reactions and promote many kinds of cell reactions. The treatment of CNS injury must take into account the various properties of cell response to effectively limit the neuronal injury after TBI [101]. By studying the different states of glial stress after TBI, and judging the stages of these stress reactions accurately, we can control the progress and deterioration of inflammation and protect the living brain tissue. For the complex stress response, it is a feasible method to accurately evaluate the stage of astrocytes and microglia by studying the ultrastructural changes. Therefore, further understanding of the ultrastructure of glial cells after TBI will be helpful to develop better methods for the treatment of TBI.

## 4. TBI-Related Ultrastructural Damage to Blood–Brain Barrier 

The blood–brain barrier (BBB) is a highly specialized, semi-permeable barrier existing between the brain and blood that serves to maintain homeostasis of the cerebral microenvironment by restricting the passage of compounds and toxins into the CNS [136,138]. Structurally, the barrier comprises an array of components including endothelial cells with tight junctions (TJ), adherens junctions, astrocytes, pericytes, and the basement membrane [138,139]. The microvascular endothelium, glial cells, pericytes and neurons, and their intercommunication and the basement membrane constitute an assembly of cells recently referred to as the “neurovascular unit” (NVU) [140]. After TBI, the loss of BBB structural integrity and heightened permeability is clearly observed. In the biopsy specimens taken from the cerebral cortex of 18 patients with brain parenchymal injury [91], a large number of dense granulations were observed, and some multivesicular bodies were observed in the cytoplasm of endothelial cells, which indicated that the pinocytic activity of endothelial cells was increased. Meanwhile, the morphology of endothelial cells changed significantly: longitudinal folds and invaginations (asterisk) were found on the surface of endothelial cells [91]. In addition, basement membrane thickening was not observed, and tight junctions between endothelial cells were intact. However, the extracellular space between endothelial cells and astrocytes appeared considerably enlarged. Moreover, it showed swelling of astrocytic perivascular processes. The cytotoxic edema of astroglial cells was formed, and the vacuolization, swollen astrocytic end-feet were also observed [91]. The electron microscopic images of brain injury sections of TBI mice by Xiaoming Yao et al. showed typical structural changes of blood–brain barrier injury (Figure 7) [90]. Recently, the results of a fluid impact damage TBI model of mice by Edwin Vázquez-Rosa et al. showed that the endothelial cells ruptured and swelled, and the number of peripheral cells decreased significantly [141].

The BBB positioned along the blood vessels of the CNS reflects the brain’s critical roles in cognition, regulating metabolism, and coordinating the functions of peripheral organs [142]. When the brain performs its complex regulatory functions, it depends on the fine control of electrical and chemical signals between neurons, so the brain needs a precise and balanced microenvironment and depends on the BBB to maintain it. The endothelial cells with tight junctions (TJ), adherens junctions, astrocytes, pericytes, and the basement membrane. Together, those components provide the structural integrity required to enable the barrier to maintain fundamental roles including supplying the brain with essential nutrients such as oxygen and glucose, mediating the efflux of waste products, and facilitating the movement of nutrients and plasma proteins [143].

In the past, BBB disruption has been found in a variety of neurologic disorders [139,144,145,146]. As for TBI, despite differences in the nature of the primary injury, loss of BBB structural integrity and heightened permeability are central features of pathogenesis [147]. The exact mechanisms by which acute CNS injury disrupts the BBB in the setting of TBI is debatable; however, acute hypertension, hyperosmolar solutions, classical inflammation, enhanced para/transcellular transport, and enhanced activity of matrix metalloproteinases (MMPs) have all been implicated, among many others [147]. Such alterations in barrier permeability following acute CNS injury arise due to loss or alterations in the function of key structural and functional components, which has major implications for injury progression and outcome [148]. The poor recovery and high mortality of TBI are largely attributable to the development of cerebral edema and elevated intracranial pressure (ICP). These serious downstream complications are closely related to the loss of structural integrity and permeability of BBB after TBI [148]. The bioenergetic crisis that ensues following TBI as a result of secondary injury processes leads to a lack of ATP production, and this leads to failure of the Na^+^/K^+^-ATPase pump, essential for the maintenance of ion homeostasis [149], which results in an inability to maintain ionic gradients across the membrane and leads to intracellular accumulation of sodium. This collapse of the ionic gradient creates an osmotic drive for water to move from the extracellular compartment to the intracellular, and such intracellular fluid accumulation leads to cellular swelling and ultimately cell rupture, causing inflammation and collateral damage to adjacent cells [94,150]. The characteristics of the endothelial cells mainly determine the BBB’s properties, but they are also affected by their communication with other NVU components [151]. Astrocytes are key cellular support of BBB integrity, and it interacts with endothelial cells through their end-feet projections that encircle the abluminal side of cerebral capillaries [94]. Such interactions are important in synchronizing metabolite levels with cerebral blood flow and vasodilation and regulating brain water content [144]. For example, the most abundant water channel protein, aquaporin 4, is predominantly expressed in astrocytic end-feet surrounding CNS vessels [152]. 

Within the NVU, the pericytes are uniquely positioned between the neurons, astrocytes, and endothelial cells [153]. Brain pericytes interact with endothelial cells and astrocytes, playing an important role of signaling in execute diverse functional responses such as regulation of blood–brain barrier permeability, angiogenesis, clearance of toxic metabolites, capillary hemodynamic responses, neuroinflammation, which are critical for brain functions in health and disease [154]. The pericyte-endothelial interactions play an important role in the maintenance of BBB with critical effects on the structure and function of the basement membrane and endothelial tight junction [154]. Pericytes also play an important role in the maturation of the BBB by guiding astrocytic end-feet to the endothelium [155]. Pericyte loss is one of the hallmarks of BBB dysfunction and has been considered as a trigger of several pathologic conditions such as abnormal BBB leakage, edema, micro-aneurysm formation, and ischemia [156]. The fact that pericyte-deficient mice have an increase in permeability of the blood–spinal cord barrier even in the absence of injury and arteriovenous malformations of the vasculature show a reduction of pericyte markers, along with extravasation of blood markers in clinical patients, support the significant importance of pericytes in maintaining endothelial barrier function [157,158].

## 5. TBI-Induced Ferroptosis

Ferroptosis, first reported in 2012 [159], is a form of regulated cell death that is different from apoptosis and is dependent on intracellular iron and lipid reactive oxygen species (ROS) [160]. Ferroptosis is primarily characterized by condensed mitochondrial membrane densities in injured cells under electron microscopy [161,162]. In two studies on a mouse model of intracerebral hemorrhage, through quantitative ultrastructural analysis of a large number of injured neurons, researchers have confirmed the existence of ferroptosis in the hemorrhagic brain [163,164]. In a recent study that employed the CCI model of TBI in mice, Xie BS et al. observed massive ferroptotic cell death within the injured area [165]. The ferroptosis was associated with the high levels of intracellular iron, lipid ROS and smaller than normal mitochondria [165,166,167]. The electron microscopic observations were consistent with those from the intracerebral hemorrhage model, showing characteristic mitochondrial atrophy (shrinkage) [164]. 

The biochemical feature of ferroptosis is an obvious accumulation of iron and toxic lipid ROS within the damaged cells. The significant morphologic phenotype at the ultrastructural level is shrinkage of mitochondria, with increased membrane density and disruption of the cristae [159,161,162,168,169]. The existence of atrophic mitochondria is the only gold standard to identify ferroptosis by transmission electron microscope [164]. The accumulation of iron-dependent toxic lipid ROS is a typical biochemical feature of ferroptosis, which results in irreparable lipid damage and membrane permeabilization [108,170]. In general, ferroptosis can be triggered when toxic levels of lipid ROS disrupt the cell’s antioxidant system [170,171]. Erastin and RSL3 are commonly used triggers of ferroptosis in cancer cells [160]. The dysfunction of lipid hydroperoxidase glutathione peroxidase 4 (GPx4) is considered to be the key event leading to ferroptosis. When the oxidation of membrane polyunsaturated fatty acids (PUFA) is unregulated due to the inactivation of GPx4, ferroptosis will occur [165,168,172]. In this case, large amounts of free radicals will be produced, leading to catastrophic lipid membrane damage [170]. Excessive iron accumulation in tissues or cells can lead to tissue damage and/or cell death [173]. However, it is not totally clear how intracellular iron affects lipid peroxidation and induces ferroptosis [170]. In view of the lack of reliable and specific biochemical markers of ferroptosis available on the market, blockage of ferroptotic cell death by lipid peroxidation inhibitors might be the easiest way to suggest the presence of ferroptosis in vivo [161,162,174]. Quantitative electron microscopic analysis of mitochondrial surface area is needed for confirmation. As a morphologic and pathognomonic feature of ferroptosis, the exact causes of mitochondrial atrophy and cristae disruption are unclear. Further research is needed.

Ferroptosis is an event that has been found to be closely related to different types of brain injury in recent years [161,162,166,175]. Further study on the mechanism of ferroptosis will help us better understand the complex pathologic grid of TBI. In addition, in view of the current status of the lack of effective treatment for TBI, we can develop specific ferroptosis inhibitors to be used in clinical trials, which might be a potential drug development direction.

## 6. Outlook

After a large number of experimental observations, we have understood that TBI is accompanied by a series of ultrastructural changes in subcellular structures. So far, various subcellular ultrastructural changes have been identified in various brain cells under electron microscopy, including different parts of the neurons (cell body, axon, and synapses), glial cells (microglia and astrocytes), and endothelial cells (Table 1). The series of ultrastructural changes after TBI show the damage to brain cells at the subcellular level. Analysis of these ultrastructural changes promotes our understanding of the pathogenesis of TBI. The causes of these different structural changes are very complex. At present, research in this area is very limited. Clarifying the molecular mechanisms that underlie the ultrastructural changes after TBI will be challenging. It can be predicted that further understanding the pathomechanisms of TBI by connecting the molecular pathology with the associated ultrastructural changes will help develop therapeutic strategies for TBI. At present, a combination of computed tomography, magnetic resonance imaging (MRI), and transcranial Doppler has been used as the diagnostic tool for TBI in clinic. However, these commonly used neuroimaging techniques provide little to no information on the key steps of secondary injury, such as excitotoxicity, neuroinflammation, cell death, proliferation, and repair [176]. In this regard, electron microscopy is an excellent tool to understand the ultrastructural changes resulting from the above secondary injury cascades. Taking advantage of the brain tissue from TBI patients would help to close the gap between preclinical and clinical studies.

Recently, studies have shown that amide proton transfer-weighted MRI (APTw-MRI) can accurately detect cerebral ischemia, hemorrhage, and inflammatory reaction after CCI and can differentiate acute intracerebral hemorrhage from ischemia in rats and in humans [132,177,178,179]. In recent years, many promising TBI drug candidates have been reported, such as melatonin receptor agonist ramelteon, which exhibits neuroprotective effects via the Nrf2 signaling pathway [180]; human wharton jelly-derived umbilical cord mesenchymal stem cells, which protects against TBI via the stem cell-related mechanism [181]; and Chinese herbal medicine rhizoma drynariae, which reduces the degree of brain injury via downregulation of immune response [182]. However, there have been no drugs convincingly reported that can protect against the ultrastructural damage or repair the damaged subcellular ultrastructure. Considering the disappointing clinical trial results for some signaling pathway target drugs, it is a potential research direction to develop drugs that can protect against or mitigate both histologic and cell ultrastructural damage when sufficient information is obtained from the human TBI brain with electron microscopy.

## Figures and Tables

**Figure 1 cells-10-01009-f001:**
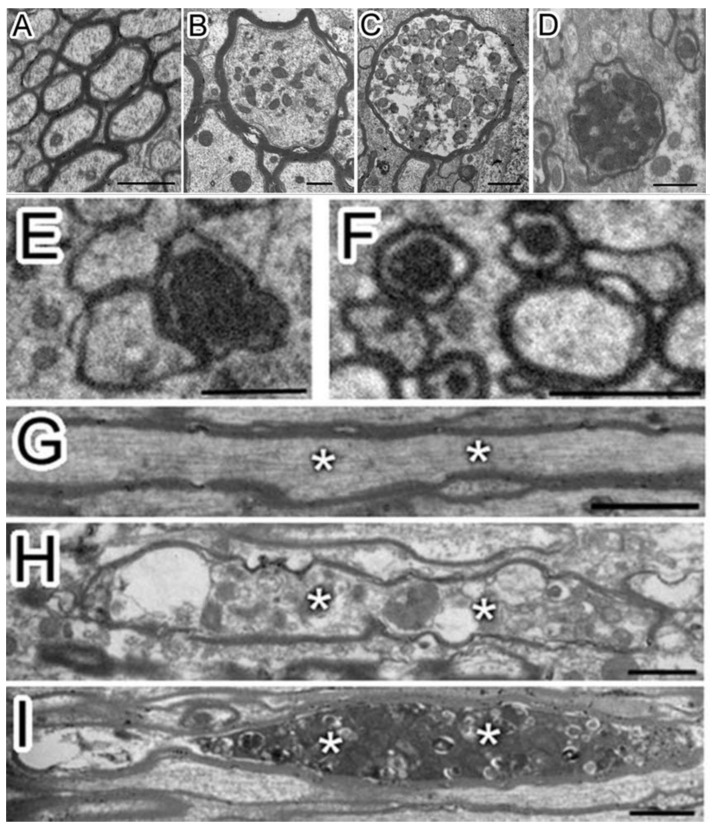
Multiple axonal degenerations were observed after traumatic brain injury (TBI) (**A**–**I**). After TBI, the damaged axons showed multi-stages degenerations from the transverse section (A, sham; B-F, TBI) and the longitudinal section (**G**, sham; **H** and **I**, TBI). The axon of sham (**A**) has a uniform cytoskeleton structure. After TBI, multiple axonal degenerations, including swelling and different densities of vesicles (**B**–**D**), were observed. Axons with very dense cytoplasm (**E**) and swollen mitochondria (**F**) were also observed, which were also considered as degenerative. Besides, the cytoskeleton structure of the sagittal section of the axon of sham (**G**, asterisk “*”) was uniform while vesicles accumulated in degenerated axons after TBI (H and I, asterisk “*”). Permission was obtained from the Oxford university press (Amanda J Mierzwa et al.) [20].

**Figure 2 cells-10-01009-f002:**
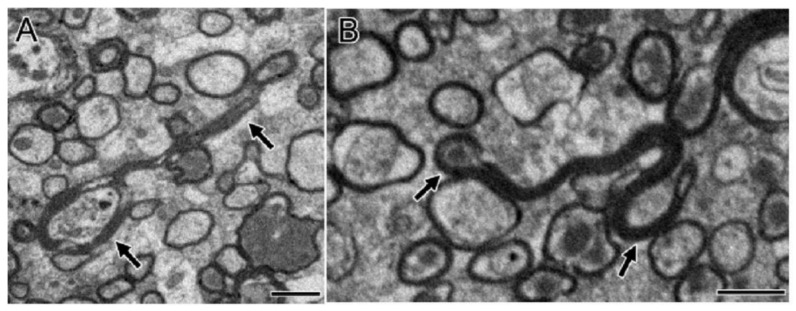
Excessive myelination correlated with axonal degeneration (**A**,**B**). Long myelin sheath (arrow) with intact axons (**A**) and degenerated axons (**B**) were observed clearly. The sagittal section of the corpus callosum shows the elongated myelin sheath folding back rather than tightly encircling the axon. Permission obtained from the Oxford university press (Amanda J Mierzwa et al.) [20].

**Figure 3 cells-10-01009-f003:**
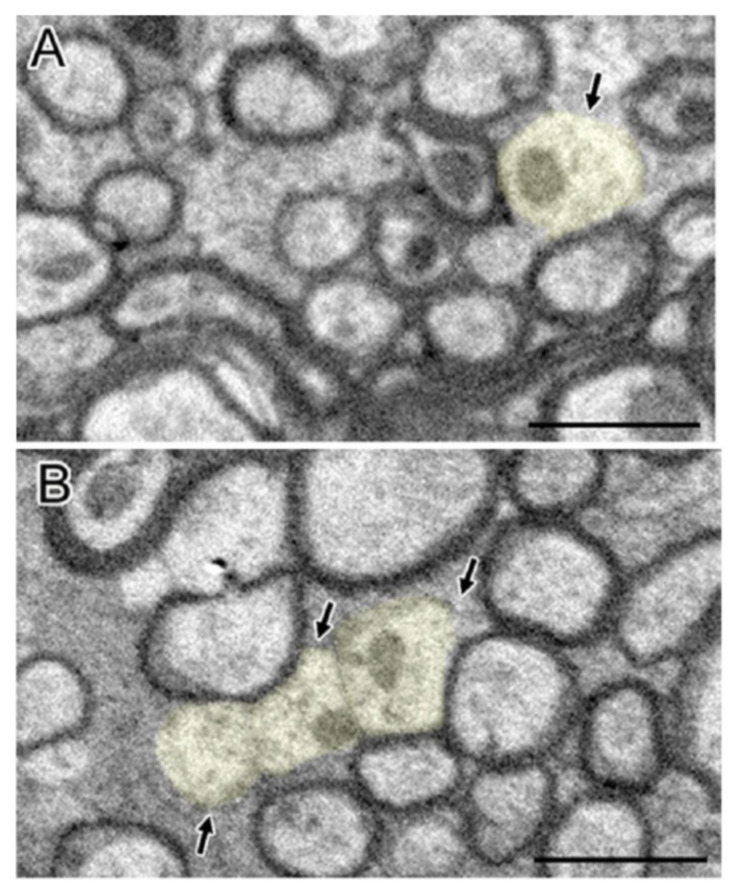
Clear and significant demyelination of intact axons after TBI were observed (**A**,**B**). Demyelinating axons (light yellow) showed having normal mitochondria and cytoskeleton structure but lack myelination. Permission obtained from the Oxford university press (Amanda J Mierzwa et al.) [20].

**Figure 4 cells-10-01009-f004:**
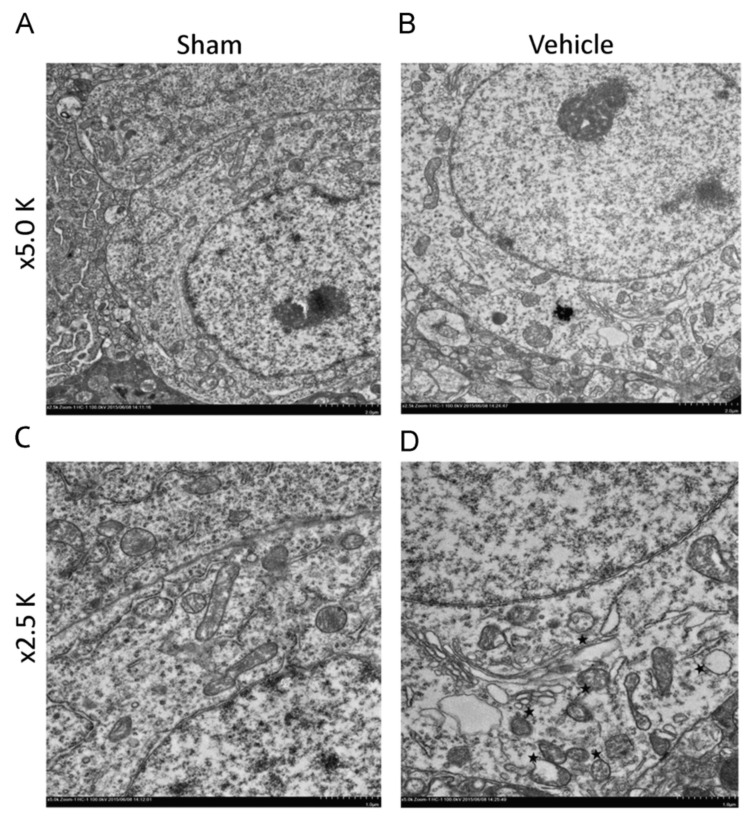
Typical electron microscopic images of mitochondrial ultrastructure of normal and damaged neurons. (**A**,**C**) showed normal morphology of mitochondria; (**B**,**D**) showed mitochondrial fragmentation of damaged neurons (Asterisk, mitochondrial fragment). Permission obtained from Elsevier (Wu, Q. et al.) [35].

**Figure 5 cells-10-01009-f005:**
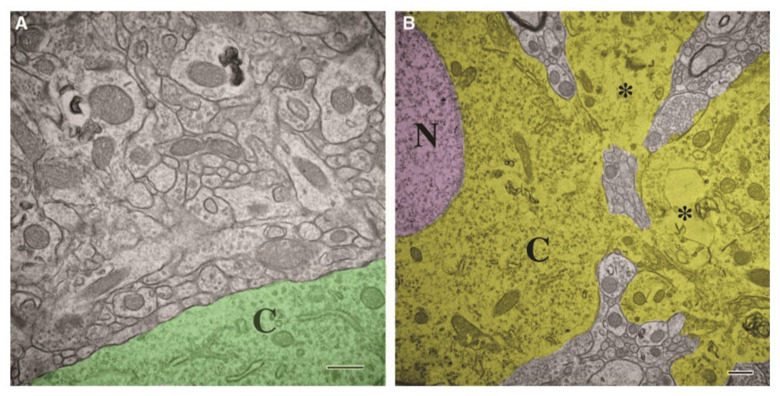
The damaged neurons showed abnormal loss and destruction of organelles which were different from normal neurons. (**A**) Ultrastructure shows remarkable preservation of organelles, and there was no obvious expansion of extracellular space in non-damaged areas. Abundant closely opposed synapses with normal complement of cytoplasmic (cytoplasm colored green) organelles and mitochondria were also observed. (**B**) Damaged areas show abnormal neurons with nucleus (N, colored pink). The cytoplasm (C, colored yellow) showed abnormal loss and destruction of organelle and most distinct filaments, tubules, or rough endoplasmic reticulum cannot be found. It seems that dilated neurotic processes (*, dilated neurotic processes) extending from the neuronal surface also were organelle depleted. Abbreviation: N, nucleus; C, cytoplasm. Permission obtained from the Mary Ann Liebert, Inc (Wiley, C. A. et al.) [1].

**Figure 6 cells-10-01009-f006:**
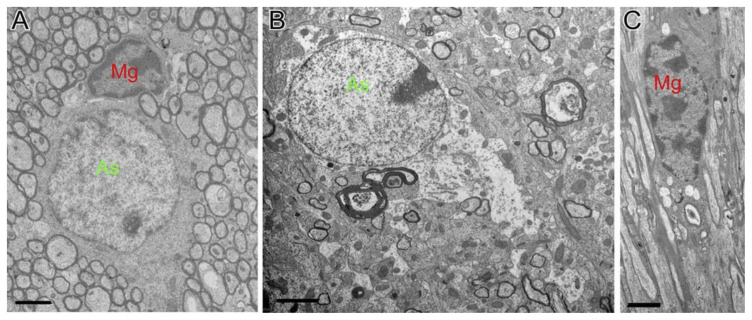
Astrocytes showing hypertrophy and prolonged protrusion were observed near injury lesions after TBI. Those astrocytes in TBI mice show hypertrophy (B, green-labeled) and microglia (**C**, red labeled) usually contained fragments while normal astrocytes (As, green-labeled) and microglia (Mg, red labeled) in sham mice (**A**) were clearly different from them. Abbreviation: As, astrocytes; Mg, microglia. Permission obtained from the Oxford university Press (Amanda J Mierzwa et al.) [20].

**Figure 7 cells-10-01009-f007:**
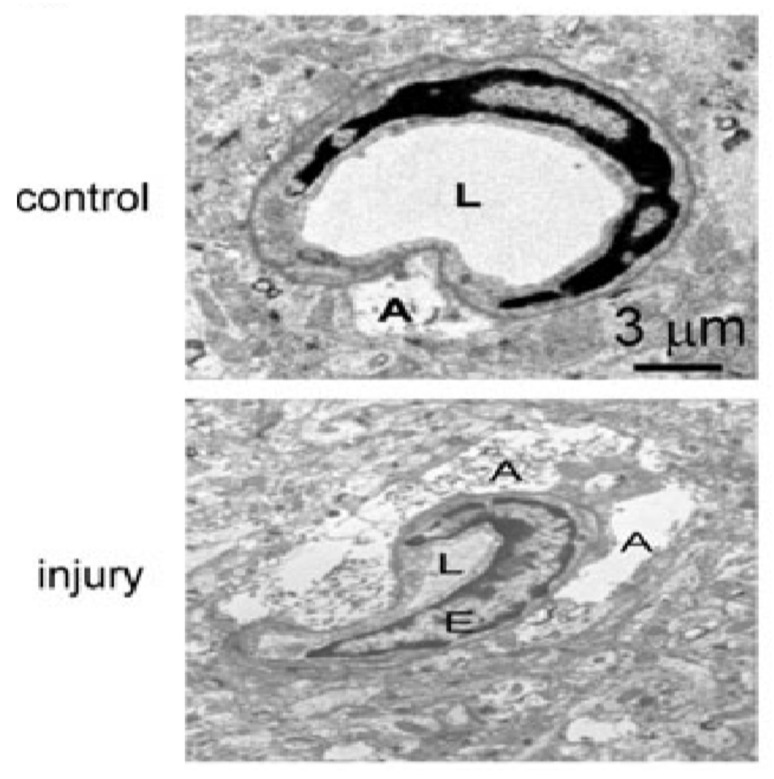
Compression of capillary lumen size and swelling of adjacent astrocytes were observed clearly after TBI. Abbreviation: A, astrocyte end foot; E, endothelial cell; L, capillary lumen. Permission obtained from Mary Ann Liebert, Inc (Yao, X. et al.) [90].

**Table 1 cells-10-01009-t001:** Ultrastructural changes after TBI.

Cell	Ultrastructural	After TBI	Reference
Neuron	Cell body	Swelling (or hydropic disintegration)	[1]
	Cell membrane	Fracture	[1]
	Nuclear	Heterochromatin loss	[1]
	Organelle	Abnormality; Loss	[1,19]
	Mitochondrion	Fracture; Swelling; Membrane rupture; Atrophy (Ferroptosis); Crista collapse and disorder; Mitochondrial density decreased	[1,19,21,22,34,35,164]
	Endoplasmic reticulum	Swelling; Dissolution	[1,56]
	Cytoskeleton	Broken; Local loss	[1,19]
	Process	Hydropic disintegration	[1]
	Axon	Destruction; Demyelination and myelination	[1,20,21,22]
Astrocyte	Cell body	Swelling; Prolonged protrusion	[20,88,89]
	Foot process	Swelling; Vacuolization	[90,91]
Microglia	Cytoplasm	Myelin fragment appeared; Cell body hypertrophy; Elongated processes (forming a hexagonal honeycomb structure)	[115,116,117,118]
Endothelia	Cytoplasm	Dense granulation and multivesical body appeared	[91]
	Surface	Longitudinal folds; invagination appeared; end-feet swelling	[90,91]

## Data Availability

Not applicable.

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
