# Peer review of "Traumatic Brain Injury: Ultrastructural Features in Neuronal Ferroptosis, Glial Cell Activation and Polarization, and Blood–Brain Barrier Breakdown"

_cells, 2021, doi:10.3390/cells10051009_

Round 1

Reviewer 1 Report

The proposed review is a nice state-of-the-art paper dedicated to TBI and its main pathological features in brain and neuromuscular spaces. 

I would suggest some minor modifications to complete the blood-brain barrier part. A better and updated definition of the BBB is required and the authors have to include a non-negligible cell type in the unit referred to as the neurovascular unit i.e. brain pericytes. As a nice reference in Cells, please take a look at Saint-Pol et al., 2020.

To complete this definition, brain pericytes have some specific features in TBI context, and the altered cross-talk between brain endothelial cells or astrocytes have been well-documented (Bhowmick et al., 2019 Exp. Neurol; Dore-Duffy et al., 2011 Neurol Research; Sakai et al., 2021 J. Pharmacol Sci.; Li W. et al., 2021 Handb Exp. Pharmacy). So an opinion and a description of brain pericyte-specific response in TBI context is expected.

Reviewer 2 Report

comments in attached file

Author Response

06- April -2021

Thank you for your letter and for your comments concerning our manuscript entitled “Traumatic brain injury: ultrastructural features in neuronal ferroptosis, glial cell activation and polarization, and blood-brain barrier breakdown”. Those comments are all valuable and very helpful for improving our paper, as well as the important guiding significance to our researches. We have studied comments carefully and have made corrections which we hope are met with approval. Revised portion are highlighted in the paper. The following is a point-by-point response to the reviewers’ comments. Major changes in the text were highlighted in yellow.

  1. Intro

Point 1: Generally, the introduction is overlong, repetitive and partly irrelevant (long discourse on ICH) Tables 1 and 2 should be removed from intro, these are summary tables which may belong in the end rather than the beginning. Optimally, the introduction can be composed of lines 27- 65 followed by lines 105-108 could be useful later in the manuscript as a summary (with references!)

Response 1: Thanks very much for your comments. We have revised accordingly based on your suggestions on pages 1 and 2. In addition, the original tables 1 and 2 have been removed to the later sections and were modified accordingly on page 19.

Point 2: line 31, necrosis is not the only mechanism of cell death after TBI, apoptosis is known to occur as well. Just using cell death instead of cell necrosis would be better here.

Response 2: As suggested, we have made changes accordingly on page 1 and line 31.

Point 3: line 41, “in most cases, recovery is poor” – this statement needs to be qualified and referenced. It is certainly not true across the spectrum of TBI severity and subjects.

Response 3: Thanks very much for your comments. We have made changes and have replaced it with new statements with reference on page 1 and line 41-43. The new statement is “It is estimated that TBI is going to be the 4th leading cause of disability-adjusted life years in 2030. TBI represents about 30–40% of all injury-related deaths across all ages, and we expect the same trend in disability rate until 2030. (Janak, J.C et al, 2015; Maas, A.I.R et al, 2017)”.

Point 4: line 44, improving TBI outcome, not improving TBI.

Response 4: Corrected on page 2 and line 47.

Point 5: line 54, initial, rather than main, impact.

Response 5: Corrected on page 2 and line 56.

Point 6: line 66-84 the overlong and detailed discourse on ICH is out of place. ICH is not the motivation for studying ultrastructural changes in TBI. It is not clear what the authors mean by the ICH findings bringing great convenience to the study of ICH, after all, we are interested in clinical benefit, not convenience. This sentence has to be rephrased and referenced.

Response 6: Thanks very much for your comments. As suggested, we have made changes accordingly in the Introduction on page 2 and line 62-64.

Point 7: Table 1 needs to be referenced (add a column). The “before TBI ”column is not informative and can be deleted – all it says is intact or normal which is the expectation anyway. Also, the association between the cell type and the changes does not make much sense. What cells are referred to in the first 5 rows? All cells? Or?

Response 7: Thanks very much for your comments. According to your opinion, we have revised the table 1 by deleting the "before TBI" column and adding the "Reference" column. As for the association between cell type and the changes, we classified the specific ultrastructural changes in specific cells according to the original references. The information conveyed in table 1 is mainly a summary of the ultrastructural changes in the references. In addition, the first 10 rows are all referred to neurons (based on the references). We have revised the table to convey accurate information on page 19.

Point 8: Table 2 needs to be referenced. What is meant by “contact channels”?  how does neuronal damage explain vasogenic edema? (it does not). How is myelination and ultrastructural change related to TBI? Do the authors mean remyelination? Necrosis and apoptosis are distinct mechanisms of cell death. If the authors mean brain tissue loss or atrophy, they should use this term rather than “necrosis”. Overall, this table needs total restructuring or omission, it does not explain much. E.g. The “clinical symptoms” column lists “nervous system dysfunction” and “inflammation” are hardly specific clinical symptoms – what one expects under clinical symptoms are things like headache, paralysis, paresis, memory impairment, etc etc. If the authors cannot relate changes to symptoms, they should use “outcome” or “sequelae” as the name of this column instead.

Response 8: Thanks very much for your comments. After careful consideration, we have decided to remove Table 2 from our manuscript.

  1. Data Review (sections 2-5)

Point 1: In general, the authors should mention the animal and model used for any of the results shown. “A TBI experiment (80)” is not a sufficient description. On the other hand, long and detailed description of processes such as edema do not have a place in a review of ultrastructural changes.

Response 1: Thanks very much for your comments. We have made changes accordingly on page 2 and line 78-81; page 3 and line 111-113, 121-122, 130, 136-138; page 4 and line 155-157; page 5 and line 187-188; page 7 and line 254-256, 258-261, 261-262, 266-270, 270-271, 271-273; page 9 and line 347-349, 353-354, 358-359; page 11 and line 431-434, 434-435, 435-439; page 12 and line 495-496, 498-499, 500-503; page 13 and line 504-505, 505-508; page 14 and line 570-572, 577-580, 580-582; page 16 and line 659-663, 670-672, 672-674; page 18 and line 739-742, 742-744. As requested, we have also removed the long and detailed description of edema on pages 2-3.

Point 2: Neuron should be changed to something like “TBI related ultrastructural damage to Neurons”

Response 2: Thanks very much for your comments. We have made changes accordingly “TBI related ultrastructural damage to Neurons” on page 2 and line 70.

Point 3: Glial cells should be similarly changed to “effects of TBI on glial cell ultrastructure” or similar

Response 3: Thanks very much for your comments. We have made changes accordingly “Effects of TBI on glial cell ultrastructure” on page 12 and line 493.

Point 4: Blood brain barrier – similarly change

Response 4: Thanks very much for your comments. We have made changes accordingly “TBI related ultrastructural damage to Blood brain barrier” on page 16 and line 650.

Point 5: Change to “TBI induced Ferroptosis”

Response 5: Thanks very much for your comments. We have made changes accordingly on page 18 and line 735.

Point 6: The definition of Ferroptosis (line &&&) should precede the current beginning of the section (line 765).

Response 6: Thanks very much for your comments. We have made changes accordingly on page 18 and line 736-739.